



# Quantifying the structural uncertainty of the aerosol mixing state representation in a modal model

Zhonghua Zheng[1], Matthew West[2], Lei Zhao[1,3], Po-Lun Ma[4], Xiaohong Liu[5], and Nicole Riemer[6]

[1]Department of Civil and Environmental Engineering, University of Illinois at Urbana-Champaign, Urbana, IL, USA
[2]Department of Mechanical Science and Engineering, University of Illinois at Urbana-Champaign, Urbana, IL, USA
[3]National Center for Supercomputing Applications, University of Illinois at Urbana-Champaign, Urbana, IL, USA
[4]Atmospheric Sciences and Global Change Division, Pacific Northwest National Laboratory, Richland, WA, USA
[5]Department of Atmospheric Sciences, Texas A&M University, College Station, TX, USA
[6]Department of Atmospheric Sciences, University of Illinois at Urbana-Champaign, Urbana, IL, USA

**Correspondence:** Nicole Riemer (nriemer@illinois.edu); Zhonghua Zheng (zhonghua.zheng@outlook.com)

**Abstract.** Aerosol mixing state is an important emergent property that affects aerosol radiative forcing and aerosol-cloud interactions, but it has not been easy to constrain this property globally. This study aims to verify the global distribution of aerosol mixing state represented by modal models. To quantify the aerosol mixing state, we used the aerosol mixing state indices for submicron aerosol based on the mixing of optically absorbing and non-absorbing species ($\chi_o$), the mixing of primary carbonaceous and non-primary carbonaceous species ($\chi_c$), and the mixing of hygroscopic and non-hygroscopic species ($\chi_h$). To achieve a spatiotemporal comparison, we calculated the mixing state indices using output from the Community Earth System Model with the modal MAM4 aerosol module, and compared the results with the mixing state indices from a benchmark machine-learned model trained on high-detail particle-resolved simulations from the particle-resolved stochastic aerosol model PartMC-MOSAIC. The two methods yielded very different spatial patterns of the mixing state indices. In some regions, the yearly-averaged $\chi$ value computed by the MAM4 model differed by up to 70 percentage points from the benchmark values. These errors tended to be zonally structured, with the MAM4 model predicting a more internally mixed aerosol at low latitudes, and a more externally mixed aerosol at high latitudes, compared to the benchmark. Our study quantifies potential model bias in simulating mixing state in different regions, and provides insights into potential improvements to model process representation for a more realistic simulation of aerosols.

## 1 Introduction

The direct and indirect climate effects of atmospheric aerosols greatly depend on the particles' spatial distribution in the atmosphere and their climate-relevant properties, including their hygroscopicity, optical properties, and their ability to act as cloud condensation nuclei (CCN) and ice nuclei (Boucher et al., 2013). These properties, in turn, are closely related to the aerosol mixing state (Ching et al., 2012; Cziczo et al., 2009; Fierce et al., 2016, 2017). Aerosol mixing state refers to the way in which different aerosol chemical species are distributed among and within the aerosol particles (Riemer et al., 2019). As shown in many observational field studies, atmospheric aerosols have complex mixing states (Bondy et al., 2018; Healy





et al., 2014; Lee et al., 2019; Ye et al., 2018; Yu et al., 2020), ranging between the two extremes of an "internal mixture", where the composition of all particles within the population is identical (and equal to the bulk composition of the aerosol), and an "external mixture", where each particle in a population consists of only a single species (which may be different for each particle).

This poses a unique challenge for the modeling of aerosols in Earth system models, which, for the sake of computational efficiency, represent aerosols by simplifying the true aerosol mixing state using various mixing-state-related assumptions. For example, bulk aerosol models predict the abundance of individual aerosol chemical species by tracking the species' mass concentrations, inherently treating the aerosol as external mixtures of, e.g., sulfate, black carbon, organic carbon, sea salt, and dust (Ghan et al., 2012). Univariate sectional models are able to represent size-resolved composition, but cannot resolve the diversity of the aerosol within a certain size range. For modal models, the ability to resolve mixing state depends on the definition and the placement of the modes. Different approaches for modal models have been developed, ranging from a small number of internally-mixed, non-overlapping modes (e.g., three modes in MAM3 (Liu et al., 2012) or CMAQv5.2 (USEPA, 2017)) to a larger number of modes that may overlap in a given size range and separate out different aerosol mixtures (e.g., nine modes in MADE3 (Kaiser et al., 2014) or 16 modes in MATRIX (Bauer et al., 2008)). For these multi-modal models, the aerosol processes of gas/aerosol partitioning and coagulation make it necessary to define rules for how the modes interact (Wilson et al., 2001). Condensation of secondary aerosol on a mode reserved for a pure species (e.g., black carbon or dust) requires moving mass over to a "mixed" mode when a critical mass fraction of secondary aerosol is exceeded. Transfer terms due to coagulation of particles in different modes can be calculated analytically (Binkowski and Shankar, 1995), and rules need to be defined regarding the destination mode after coagulation. Hence, the choice of the number of modes, their compositions, and the criteria for transfer between modes are user-defined, which introduces structural uncertainty in aerosol simulations that still needs to be quantified.

Given that modal models are to some extent "mixing-state-aware", the question arises: how well do modal models represent mixing state? Due to the scarcity of relevant observational data, we are not yet at the point where we can comprehensively validate model output of aerosol mixing state as is done for other aerosol-related quantities, such as bulk mass concentrations or aerosol optical depth. However, higher-detail models can serve as benchmarks to perform a verification of simulated aerosol mixing state. This paper aims to verify the global distribution of aerosol mixing state represented by a modal model by using benchmark simulations from the particle-resolved stochastic aerosol model PartMC-MOSAIC (Riemer et al., 2009; Zaveri et al., 2008). We used the aerosol mixing state index $\chi$ (Riemer and West, 2013) as a metric to quantify aerosol mixing state. The mixing state index $\chi$ can be interpreted as a label for particle populations to rigorously characterize where the population lies on the spectrum from external ($\chi = 0\%$) to internal ($\chi = 100\%$) mixture. This concept has been successfully applied to observational data (Healy et al., 2014; Ye et al., 2018) and for error quantification studies (Ching et al., 2017, 2018). Particularly relevant for this work is the study by Ching et al. (2017), which showed that assuming an internal mixture when the aerosol is actually not completely internally mixed can result in errors of up to 150% in CCN predictions. Similarly, this assumption can result in up to 120% error in the predictions of the volume absorption coefficient (Yao et al., 2021).



PartMC-MOSAIC tracks the composition of individual particles and therefore resolves aerosol mixing state explicitly (Riemer et al., 2009; Zaveri et al., 2008). However, this modeling approach is computationally very expensive and therefore not practical for large-scale simulations of several months or years of simulation time. To estimate the global spatial distribution of mixing state, we recently developed a machine-learned (ML) model based on high-detail particle-resolved simulations (Zheng et al., 2021) that uses inputs that are known from global model simulations to predict $\chi$. In this paper, we use this ML model to predict the spatial distribution of the mixing index $\chi$ and then compare the results with $\chi$-values that are derived from the Community Earth System Model Version 2 (CESM2 version 2.1.0; Danabasoglu et al., 2020) using the 4-mode version of the Modal Aerosol Module (MAM4; Liu et al., 2016).

This paper is organized as follows. In Section 2 we introduce the setup of the Earth system model simulations. The definition of mixing state indices and the derivation of aerosol mixing state indices for modal models are given in Section 3. Section 4 briefly describes the ML model generated with machine learning and particle-resolved modeling for estimating the benchmark aerosol mixing state indices. Section 5 focuses on the comparison of mixing state indices from the particle-resolved and modal models, and Section 6 summarizes our findings.

## 2 Global model simulations

Here we employed CESM2 to provide the global model simulation data. Specifically, we used the component set FHIST to set up the global simulations with aerosols. This component set represents a typical historical simulation in the Community Atmospheric Model (CAM6; Bogenschutz et al., 2018) using an active atmosphere and land with prescribed sea-surface temperatures and sea-ice extent, and a 1-degree finite volume dycore with the forcing data available from 1979 to 2015.

MAM4 is the default aerosol module of this component set, which represents the aerosol size distribution with four lognormal modes (Aitken, accumulation, coarse, and primary carbon modes; Liu et al., 2016). MAM4 tracks six aerosol species, and these are distributed over the four modes as follows. The Aitken mode consists of dust, sulfate, secondary organic aerosol (SOA), and sea salt. The accumulation mode includes sulfate, SOA, sea salt, primary organic matter (POM), black carbon (BC), and dust. The coarse mode contains sulfate, dust, and sea salt. The primary carbon mode contains only BC and POM, which are supplied by primary aerosol emissions.

The choice of modes in MAM4 is motivated by the desire to treat the microphysical aging of the primary carbonaceous aerosols in the atmosphere (Liu et al., 2016), similar to other modal models used in regional or global models (Riemer et al., 2003; Vogel et al., 2009). In MAM4, mass and number concentrations of BC and POM in the primary carbon mode are transferred to the accumulation mode by the processes of intermodal coagulation and condensation of SOA and sulfuric acid onto the primary carbon mode. The accumulation mode then represents aged BC and POM, as these species are internally mixed with other aerosol species. The MAM4 treatment of aging is critical for improving the long-range transport of carbonaceous aerosols to remote regions such as the polar region, which suffered from a low bias in a prior version of the model when only three internally-mixed modes were used.





We ran the model for the year 2011 with 6 years (2005-2010) of spinup. The simulation was conducted at a resolution of 0.9° latitude by 1.25° longitude along with emission inventories from CMIP6 emissions (Emmons et al., 2020). We stored the

90 instantaneous outputs every three hours during the simulation, which yields 2920 timestamps for each surface-layer grid cell for the entire year of simulation time. The surface layer was chosen to be in line with the PartMC-MOSAIC model scenarios that were used as training data for the ML models of mixing state indices (see Section 3) and which were designed to represent conditions in the planetary boundary layer.

## 3 Aerosol mixing state indices: definition and calculation

### 3.1 Particle-based aerosol mixing state index

The mixing state index $\chi$ (Riemer and West, 2013) quantifies where an aerosol population lies on the continuum from external to internal mixing. That is, how "spread out" the chemical species are over an aerosol population. We will focus here on the mixing state of sub-micron aerosols ($PM_{1.0}$) due to their relevance for light scattering and absorption (Wang et al., 2015), and their contribution to CCN formation (Asmi et al., 2011; Pierce et al., 2015; Yu and Luo, 2009).

To summarize, the mixing state index $\chi$ is given by the affine ratio of the average particle species diversity, $D_\alpha$, and bulk population species diversity, $D_\gamma$, as

$$\chi = \frac{D_\alpha - 1}{D_\gamma - 1}. \tag{1}$$

The diversities $D_\alpha$ and $D_\gamma$ are calculated as follows. First, the per-particle mixing entropies $H_i$ are determined for each particle by

$$H_i = \sum_{a=1}^{A} -p_i^a \ln p_i^a. \tag{2}$$

Here, $A$ is the number of distinct aerosol species and $p_i^a$ is the mass fraction of species $a$ in particle $i$. These values are then averaged (mass-weighted) over the entire population to obtain the average particle species diversity $D_\alpha$ by

$$H_\alpha = \sum_{i=1}^{N_p} p_i H_i, \tag{3}$$

$$D_\alpha = e^{H_\alpha}, \tag{4}$$

where $N_p$ is the total number of particles in the population and $p_i$ is the mass fraction of particle $i$ in the population. Finally, the bulk diversity $D_\gamma$ is calculated as

$$H_\gamma = \sum_{a=1}^{A} -p^a \ln p^a, \tag{5}$$

$$D_\gamma = e^{H_\gamma}, \tag{6}$$





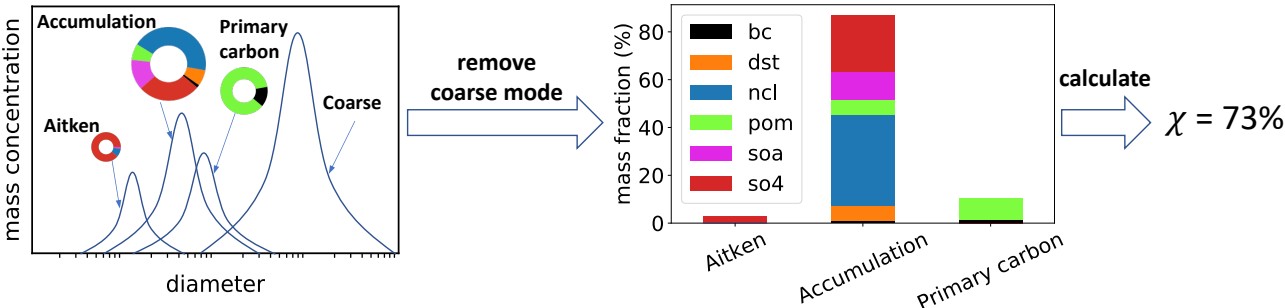

**Figure 1.** Illustration of the mode-based calculation of the aerosol mixing state index. The coarse mode is removed because only modes dominated by submicron particles are used for calculations.

where $p^a$ is the bulk mass fraction of species $a$ in the population.

Note that the definition of "species" for calculating $\chi$ is based on application needs. It can be based on operationally defined chemical species (Healy et al., 2014; Ye et al., 2018), elemental composition (Fraund et al., 2017; O'Brien et al., 2015), or species groups such as volatile and nonvolatile species (Dickau et al., 2016) or hygroscopic and non-hygroscopic species (Ching et al., 2017; Hughes et al., 2018). In this paper we consider three different definitions of $\chi$, which we explain in more detail in Section 3.2.

### 3.2 Mode-based aerosol mixing state index

The framework laid out in Section 3.1 can be easily generalized to a modal modeling framework. The bulk mixing entropy, $H_\gamma$, and the bulk diversity, $D_\gamma$, can be calculated using the bulk mass fractions, $p^a$, of species $a$ from the MAM4 simulation and Equations (5) and (6). To calculate the average particle mixing entropy, $H_\alpha$, and the average particle species diversity, $D_\alpha$, we use

$$H_m = \sum_{a=1}^{A} -p_m^a \ln p_m^a, \tag{7}$$

$$H_\alpha = \sum_{m=1}^{M} p_m H_m, \tag{8}$$

$$D_\alpha = e^{H_\alpha}, \tag{9}$$

where $p_m^a$ is the mass fraction of species $a$ in mode $m$, $p_m$ is the mass fraction of mode $m$ in the population, and $H_m$ are the per-mode mixing entropies. Finally, the mixing state index, $\chi$, can be calculated using Equation (1). Note that Eqs. (7) and (8) are analogous to Eqs. (2) and (3). A detailed derivation of these equations is provided in the Appendix A.

In this study, we consider the mixing states of submicron aerosols including the Aitken, accumulation, and primary carbon modes and we do not include the coarse mode because the coarse particles are above one micron. Since the mixing entropies





**Table 1.** Aerosol mixing state indices definitions. Six aerosol species (bc: black carbon, dst: dust, ncl: sea salt, pom: primary organic matter, soa: secondary organic aerosol, so4: sulfate) are used in calculating the aerosol mixing state indices based on different species groupings. The mixing state indices $\chi_o$, $\chi_c$, and $\chi_h$ are based on two grouped surrogate species.

| Aerosol Mixing State Index (symbol) | Grouped Species | |
| --- | --- | --- |
| optical property ($\chi_o$) | (bc) | (pom, dst, ncl, soa, so4) |
| primary carbon ($\chi_c$) | (bc, pom) | (dst, ncl, soa, so4) |
| hygroscopicity ($\chi_h$) | (bc, pom, dst) | (ncl, soa, so4) |

are mass-weighted (rather then number-weighted), the mixing state index is more representative of the modes with the larger particles, i.e., the accumulation and primary carbon modes.

## 3.3 Grouped surrogate species

Here we compare and contrast the aerosol mixing state indices defined in three different ways, namely based on the mixing of optically absorbing and non-absorbing species ($\chi_o$), based on the mixing of primary carbonaceous and non-primary carbonaceous species ($\chi_c$), and based on the mixing of hygroscopic and non-hygroscopic species ($\chi_h$). Table 1 shows the definitions of these aerosol mixing state indices.

For $\chi_o$, we considered two surrogate species: black carbon (strongly absorbing, assigned a mass absorption coefficient in CESM2 at 533 nm and 0% RH of 8.144 $\mathrm{m^2\,g^{-1}}$) and the five other aerosol species grouped together (less absorbing or non-absorbing, with mass absorption coefficients in CESM2 at 533 nm and 0% RH of 0.1442 $\mathrm{m^2\,g^{-1}}$, $9.975 \times 10^{-2}$ $\mathrm{m^2\,g^{-1}}$, $4.703 \times 10^{-2}$ $\mathrm{m^2\,g^{-1}}$, $2 \times 10^{-6}$ $\mathrm{m^2\,g^{-1}}$, $5 \times 10^{-7}$ $\mathrm{m^2\,g^{-1}}$, for POM, SOA, dust, sea salt, and sulfate, respectively). Thus, a lower value in $\chi_o$ refers to the case where the strongly absorbing species black carbon (Yang et al., 2009) and the sum of the other species (termed "non-absorbing" here for convenience) are more externally mixed.

The index $\chi_c$ is motivated by the primary carbon treatment of MAM4, where the primary particulate organic matter and black carbon are assigned to a separate primary carbon mode (Liu et al., 2016). A lower value in $\chi_c$ refers to the situation where the primary carbonaceous species and all other species exist separately in different particles.

Similarly, $\chi_h$ was also calculated from two surrogate species. We combined black carbon, primary organic matter and dust as one surrogate species, given their comparatively lower hygroscopicities (kappa values of ∼0, ∼0, and 0.068, respectively). Accordingly, NaCl (1.16), SOA (0.14), and sulfate (0.507) were grouped as the other surrogate species. Here, a lower value in $\chi_h$ represents the case where hygroscopic and non-hygroscopic species tend to be present in separate particles.

## 4 Machine-learned models of mixing state indices

Aerosol mixing state indices can be calculated directly using particle-resolved modeling, but this comes with large computational costs. Alternatively, Zheng et al. (2021) developed ML models, which integrate machine learning and particle-resolved aerosol simulations to estimate aerosol mixing state indices. To generate the training and testing data sets for developing such





ML models, an ensemble of particle-resolved model scenarios was created using the particle-resolved model PartMC-MOSAIC (Riemer et al., 2009; Zaveri et al., 2008). In brief, PartMC-MOSAIC simulates individual aerosol particles within a representative volume of air, including stochastic coagulation, particle-phase thermodynamics, gas- and particle-phase chemistries, and
160 dynamic gas-particle mass transfer. Thus, the composition of the individual particles within a population evolves dynamically and assumptions about mixing state are not necessary.

The strategy to generate the data was to vary the input parameters (45 in total) for the PartMC-MOSAIC model, including primary emissions of different aerosol types (e.g., carbonaceous aerosol and dust emissions), primary emissions of gas phase species (e.g., $SO_2$, $NO_2$, and various volatile organic compounds), and meteorological parameters (see Table 1 in Zheng et al.
(2021) for more information). For instance, to vary the gas emissions, scaling factors were sampled from 0% to 200% for different gas species, based on the emission rates in Riemer et al. (2009). A Latin hypercube sampling approach was employed to sample the parameter space efficiently for the training and testing data sets.

The ML models were derived by the machine learning algorithm eXtreme Gradient Boosting (XGBoost; Chen and Guestrin, 2016) from 45 000 particle populations. Each ML model was a tree-based ensemble model that could handle complex non-
170 linear interactions and collinearity among features. The hyperparameters were determined by grid search with 10-fold cross-validation. The ML models can be expressed as

$$\chi_S(x,y,t) = f_S\big(A(x,y,t), G(x,y,t), E(x,y,t)\big), \tag{10}$$

where $\chi_S(x,y,t)$ is the mixing state index ($\chi_o$, $\chi_c$, or $\chi_h$) at location $(x,y)$ in the model layer nearest the surface at time $t$ and $f_S$ denotes the function for calculating the corresponding mixing state index $\chi_S$. The set names $A$ (aerosol), $G$ (gas),
and $E$ (environmental) represent the predictors (features) used for predicting the mixing state index. Aerosol species include black carbon, mineral dust, sea salt, primary organic aerosol, secondary organic aerosol, and sulfate. Of note is that we used the bulk (not the per-mode) concentrations of submicron aerosol species as the features. The gas species include dimethyl sulfide, hydrogen peroxide, sulfuric acid, ozone, semi-volatile organic gas and sulfur dioxide. The environmental variables are air temperature, relative humidity, and solar zenith angle. Table 2 shows the performance of the ML models when predicting
the mixing state indices. The mixing state calculation in this study was purely based on the above six aerosol species (excluding other aerosol species) for a fair comparison with the mode-based aerosol mixing state index, which resulted in slightly different performance of the ML model compared to Zheng et al. (2021). The average error of the ML model is about 5% for $\chi_o$ and 8% for $\chi_c$ and $\chi_h$ (measured by mean absolute error).

## 5 Results

### 5.1 Quantitative comparison of mode-based and particle-based mixing state indices

Let $\chi_{S,t}^{ML}$, and $\chi_{S,t}^{MAM4}$ denote the mixing state indices computed by the ML model and by the MAM4 model for each grid cell at timestamp $t$, respectively. The corresponding time-averaged values for a certain time interval and for each grid cell are $\overline{\chi_S^{ML}}$



**Table 2.** Predictive performance of the ML models using the testing data set. Metrics include the mean absolute error (MAE), root-mean-square error (RMSE), median absolute deviation (MAD), index of agreement ($d$; Willmott, 1981), Pearson correlation coefficient (PCC), and coefficient of determination ($r^2$).

| $\chi$ | MAE | RMSE | MAD | $d$ | PCC | $r^2$ |
|---|---|---|---|---|---|---|
| | | | XGBoost ML Models | | | |
| $\chi_\mathrm{o}$ | 0.048 | 0.072 | 0.030 | 0.974 | 0.953 | 0.906 |
| $\chi_\mathrm{c}$ | 0.079 | 0.107 | 0.056 | 0.955 | 0.916 | 0.836 |
| $\chi_\mathrm{h}$ | 0.082 | 0.112 | 0.057 | 0.955 | 0.916 | 0.835 |

and $\overline{\chi_S^{\mathrm{MAM4}}}$. Here we consider the full year as the time-averaging interval. An analysis of the seasonal variation of mixing state indices can be found in Zheng et al. (2021).

To compare the annual mean values, we calculated the mean difference ($\overline{\Delta\chi_S}$) and the mean absolute difference ($\overline{|\Delta\chi_S|}$) for each grid cell of the layer closest to the surface:

$$\overline{\Delta\chi_S} = \frac{1}{T}\sum_{t=1}^{T}(\chi_{S,t}^{\mathrm{MAM4}} - \chi_{S,t}^{\mathrm{ML}}) = \overline{\chi_S^{\mathrm{MAM4}}} - \overline{\chi_S^{\mathrm{ML}}}, \tag{11}$$

$$\overline{|\Delta\chi_S|} = \frac{1}{T}\sum_{t=1}^{T}(|\chi_{S,t}^{\mathrm{MAM4}} - \chi_{S,t}^{\mathrm{ML}}|), \tag{12}$$

where the subscript $S$ refers to the mixing state index (o, c, or h), and the total number of timestamps is $T = 2920$. Since it only makes sense to quantify mixing state when at least two species are present in a given location, areas where the mass fraction of any one surrogate species was higher than 99% for $\chi_\mathrm{o}$ (due to the low mass fraction of black carbon) and 97.5% for $\chi_\mathrm{c}$ and $\chi_\mathrm{h}$ were ignored for the calculation and appear as hatched areas in Figure 3. We will first discuss the overall probability density functions of these quantities (Figure 2) and then their spatial distributions (Figure 3).

Figure 2 shows the probability density functions of the annual averaged mixing state indices computed by the ML model ($\overline{\chi_S^{\mathrm{ML}}}$), by MAM4 ($\overline{\chi_S^{\mathrm{MAM4}}}$), their average difference ($\overline{\Delta\chi_S}$), and their average absolute difference ($\overline{|\Delta\chi_S|}$) for each surface-layer grid cell. The results show large discrepancies in mixing state indices between the ML model and the MAM4 model, without a clear relationship between them (see Figure 2d–f).

The annual average of the mixing state index $\chi_\mathrm{o}$ estimated by the ML model, $\overline{\chi_\mathrm{o}^{\mathrm{ML}}}$, ranged between 55% and 96%, with a mean of 73%. Calculated by the MAM4 model, $\overline{\chi_\mathrm{o}^{\mathrm{MAM4}}}$ varied spatially from 46% to 99.76%, with a higher mean of 86%. The similar mean values of $\overline{\Delta\chi_\mathrm{o}}$ (14%) and $\overline{|\Delta\chi_\mathrm{o}|}$ (18%) were caused by higher values in $\chi_\mathrm{o}^{\mathrm{MAM4}}$ compared to $\chi_\mathrm{o}^{\mathrm{ML}}$, which is confirmed below with Figure 3. The averaged mixing state index $\overline{\chi_\mathrm{c}^{\mathrm{ML}}}$ ranged between 31% and 84% with a mean of 54%, while $\overline{\chi_\mathrm{c}^{\mathrm{MAM4}}}$ had a wider range (from 9% to 99.81%) with a mean (of 58%). Similarly, $\overline{\chi_\mathrm{h}^{\mathrm{ML}}}$ ranged from 21% to 81% with a mean of 58%, while $\overline{\chi_\mathrm{h}^{\mathrm{MAM4}}}$ varied between 10% to 99.85% with a mean of 63%. The large discrepancy between the mean difference (4.8% for $\overline{\Delta\chi_\mathrm{c}}$ and 4.7% for $\overline{\Delta\chi_\mathrm{h}}$) and mean absolute difference (30% for $\overline{|\Delta\chi_\mathrm{c}|}$ and 38% for $\overline{|\Delta\chi_\mathrm{h}|}$) indicates that




the errors in $\chi_c$ and $\chi_h$ were symmetric (positive and negative) but large. The maximal errors in $\overline{|\Delta\chi_c|}$ and $\overline{|\Delta\chi_h|}$ between the two methods was up to 59 and 76 percentage points, respectively.

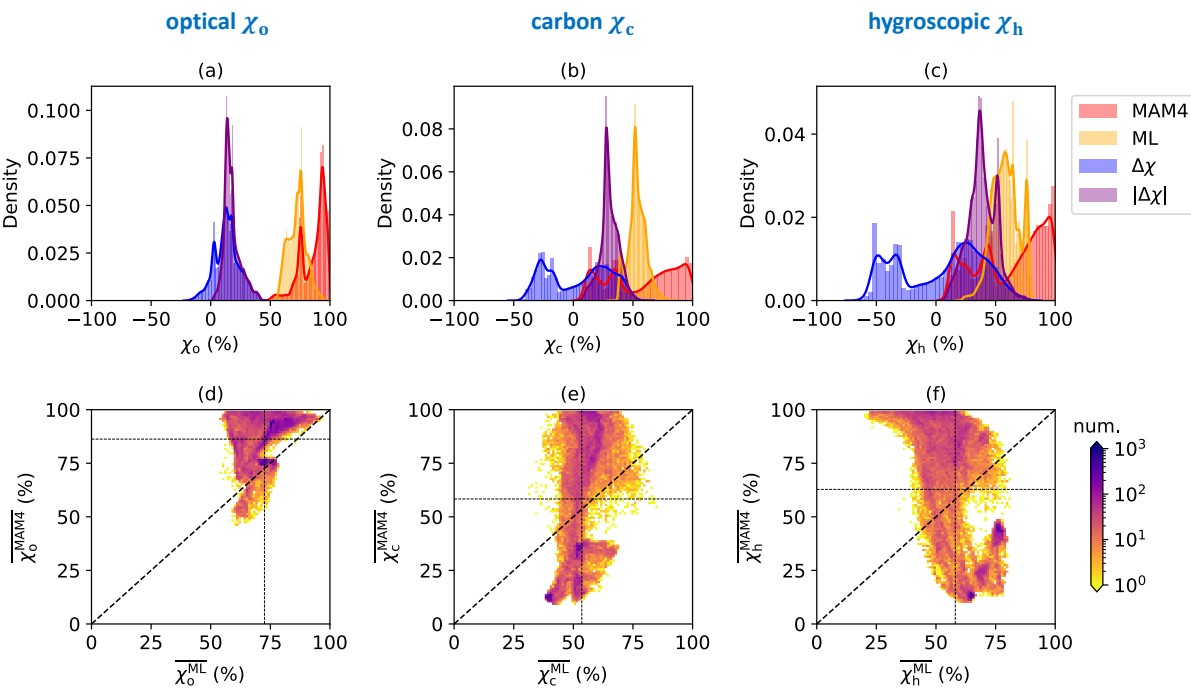

**Figure 2.** Probability density functions of annual averaged mixing state indices using the MAM4 model and ML model. The thin black lines refer to their mean values.

The implications of these discrepancies are more easily discussed with Figure 3, which illustrates the global spatial distribution of annually-averaged mixing state indices predicted by the ML model (first column), MAM4 (second column), their mean difference (third column), and their mean absolute difference (fourth column). The differences in mixing state indices between
215 the ML model and MAM4 varied strongly across the globe.

High values of $\overline{\chi_o^{ML}}$ occurred in the continental regions (77%) compared to oceans (69%). Specifically, the ML model predicted high values for $\chi_o$ in Central Africa (20°S–15°N, 12–30°E), the Arctic (66.5–90°N), and Southern Asia (5–38°N, 60–90°E). These are also the regions with relatively larger mass fractions of black carbon (~5%, see Fig. A2). The mixing state index $\chi_o^{MAM4}$ showed a higher degree of internal mixing over the globe (with a median of 90%) compared to the ML model.
The only exceptions were oceans in the Northern hemisphere at the mid-latitudes (45–60°N, dominated by sea salt, sulfate, and secondary organic aerosol in the accumulation mode) and Antarctica (66.5–90°S, dominated by sea salt and sulfate in the accumulation mode as well as sulfate in Aitken mode), where $\overline{\chi_o^{MAM4}}$ was 75%. Qualitatively, the MAM4 model captured the


trend that areas with high black carbon concentration (defined here as concentrations above the 95% percentile) tended to have higher $\chi_o$-values.

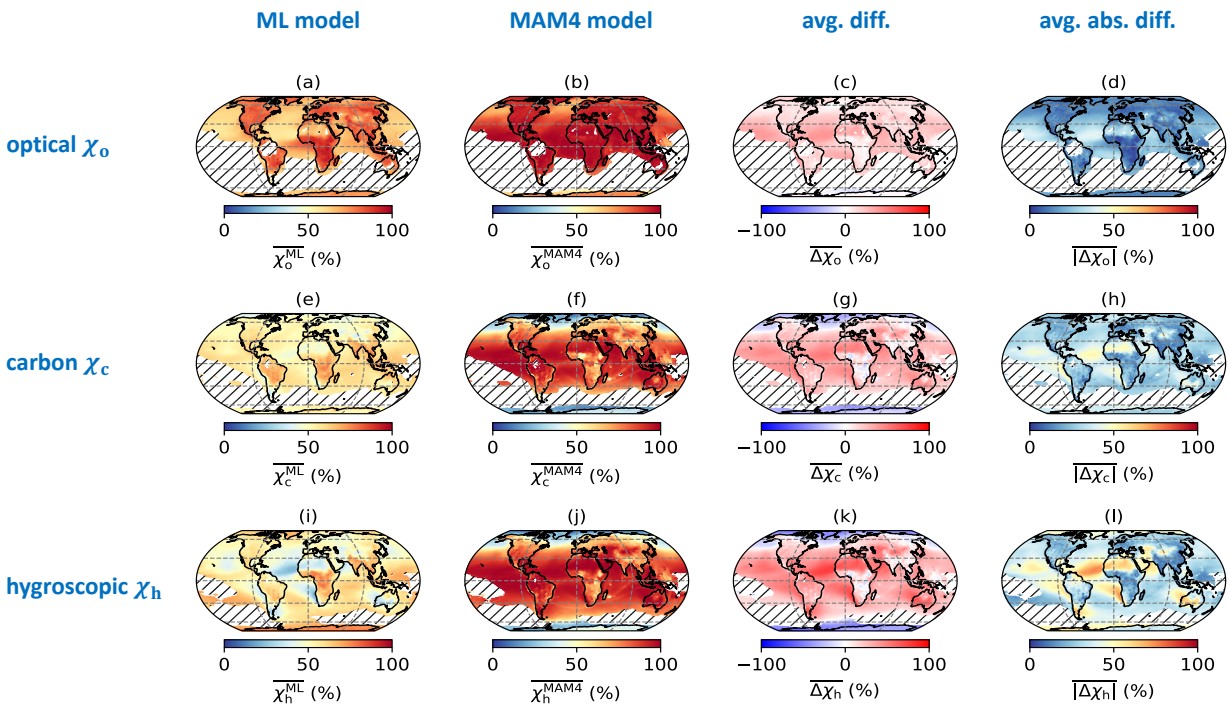

**Figure 3.** Global distribution of annually averaged mixing state indices ($\chi_o$, $\chi_c$, and $\chi_h$) using the ML model, MAM4 model, their mean difference ($\overline{\Delta\chi}$), and mean absolute difference ($\overline{|\Delta\chi|}$). Areas are hatched where the mass fraction of any one surrogate species was higher than 99% for $\chi_o$ (due to the low mass fraction of black carbon) and 97.5% for $\chi_c$ and $\chi_h$.

The ML model estimate $\overline{\chi_c^{\mathrm{ML}}}$, suggested a rather homogeneous spatial distribution of the annually averaged mixing state, with values of approximately 50%. Compared to $\overline{\chi_c^{\mathrm{ML}}}$, $\overline{\chi_c^{\mathrm{MAM4}}}$ values were lower (primary carbonaceous aerosol more externally mixed) at high latitudes, and higher at low and mid latitudes (primary carbonaceous aerosol more internally mixed). Note that, while $\overline{\chi_c^{\mathrm{MAM4}}}$ values were similar in the Arctic and Antarctic, the abundance of primary carbonaceous species was predicted to be higher in the Arctic compared to the Antarctic (see Fig. A2).

The spatial distributions of $\overline{\chi_h^{\mathrm{MAM4}}}$ were similar to $\overline{\chi_c^{\mathrm{MAM4}}}$. That is, the MAM4 model predicted that the hygroscopic species and non-hygroscopic species were more externally mixed at high latitudes and more internally mixed at low latitudes. In contrast, the spatial distribution of $\overline{\chi_h^{\mathrm{ML}}}$ shows qualitative differences compared to $\overline{\chi_c^{\mathrm{ML}}}$ in two aspects. First, $\overline{\chi_h^{\mathrm{ML}}}$ was higher than $\overline{\chi_c^{\mathrm{ML}}}$ at high latitudes, meaning that hygroscopic species and non-hygroscopic appeared more internally mixed than primary carbonaceous and non-carbonaceous species in this region. Second, areas over the North Atlantic Ocean (0–20°N, 20–45°W),




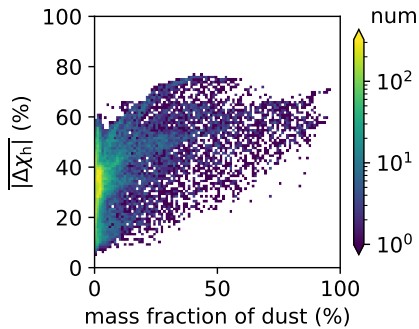

**Figure 4.** Dependence of mean absolute difference of $\chi_h$ on dust mass fractions for all model gridpoints.

Southern Africa (5–32°S, 5–20°E), and Australia (10–30°S, 100–140°E) appeared rather externally mixed. These are areas where mineral dust is the dominant aerosol species (see Fig. A2).

These two facts lead to the overall finding that $\chi_h$ exhibits the largest differences between the two methods. This applies especially to regions where mineral dust was the dominant aerosol species, which points to an important structural issue of the four-mode setup used in MAM4. While the ML model predicted a more external mixture in these regions (dust externally mixed from sea salt and other species), the MAM4 model could not represent this because the accumulation mode included all six aerosol species in an internal mixture. Figure 4 illustrates the relationship of the mean absolute difference of $\chi_h$ and the mass fraction of dust for all model gridpoints. It confirms that gridpoints with large dust mass fractions were associated with larger mean absolute differences in $\chi_h$. These results confirm the tradeoff discussed in Liu et al. (2012): MAM3 (and MAM4 in Liu et al. (2016)) intentionally combines dust and sea salt in the same mode to reduce the computational burden, however this simplification does not always realistically reflect the aerosol mixing state in the ambient atmosphere.

It is interesting to note that the areas where sea salt is present, but not dust, are *not* associated with large errors, even though sea salt—just like mineral dust—is a primary aerosol type. The reason for this lies in our surrogate species definitions (Table 1) for computing the mixing state index. Sea salt is always grouped with at least two other species (e.g., SOA and sulfate for $\chi_h$). Therefore, none of the mixing state indices as defined here tell us how externally mixed sea salt is when it is considered as a single aerosol type.

Figure 5 further demonstrates the zonal mean annual aerosol mixing state indices, highlighting that differences between $\chi_c$ and $\chi_h$ tended to be zonally structured, where the MAM4 model overestimated at low latitudes, while it underestimated at high latitudes relative to the ML model. In contrast, the MAM4 model overestimated $\chi_o$ at all latitudes north of 60°S.

## 5.2 Interpretation of findings

From Section 5.1, the following picture emerges: MAM4 overestimates the mixing state index $\chi_o$ except in regions at high latitudes in the Southern hemisphere. At the same time, $\chi_c$ and $\chi_h$ are overestimated at low to mid-latitudes and underestimated



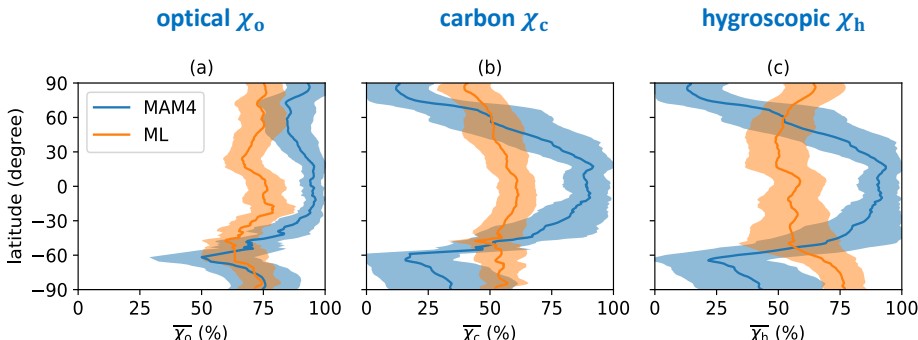

**Figure 5.** Zonal mean annual aerosol mixing state indices (a) $\chi_o$, (b) $\chi_c$, and (c) $\chi_h$ using the MAM4 model and ML model. The bands refer to the standard deviation.

at high latitudes. These findings point towards a too-rapid transfer from the carbonaceous mode to the accumulation mode at low to mid latitudes, and a too-slow transfer at high latitudes.

To conceptually illustrate these relationships, here we use $\chi_o$ and $\chi_c$ as examples and contrast the conditions for high and low latitudes. Figure 6a–f shows conditions representative of high latitudes. A grid cell sampled from the CESM2/MAM4 simulation (73°N, 151° W) contains 15% BC and 37% POM, distributed over the accumulation and primary carbon mode as shown in Figure 6a and d. The corresponding value for $\chi_o$ is 80%. Figure 6b depicts particle population that was sampled from the MAM4 population in Figure 6a. All particles, except for the smallest ones (corresponding to Aitken mode particles), contain BC, which results in the relatively high mixing state index value for $\chi_o$. Note that in MAM4 BC is not included in the Aitken mode by definition. Considering the same particle population, but now evaluating the mixing state metric $\chi_c$, which quantifies the degree of mixing of primary carbon and other species, yields the following observation. The entire primary carbon mode, by definition, consists of POM and BC, which results in an appreciable number of particles that contain only primary carbon (BC+POM), giving a mixing state index $\chi_c$ of only 27%.

We now compare the MAM4-sampled particle populations above to particle populations that were sampled from our PartMC scenario library. We searched for populations with similar mass fractions of BC and POM as in the MAM4 populations and that were simulated at a similiar latitude as the grid point location of the CESM2/MAM4 model output. Figure 6c shows that the PartMC results have comparatively more BC-free particles, and Figure 6f shows that comparatively more particles are mixtures of primary carbon and other species. Overall, this means that in MAM4 BC appears too internally mixed (because irrespective of whether BC is placed in the primary carbon or accumulation mode, it is by design mixed with other species) and that at high latitudes the primary carbon mode is not transferring mass to the accumulation mode as quickly as is the case in PartMC simulations.

The reason why MAM4 behaves in this way can be explained by the aging process treatment in MAM4. Aging in MAM4 is formulated using a threshold criteria. That is, BC and POA mass is transferred from the primary carbon mode to the accu-



mulation mode when a certain threshold of sulfate and SOA has condensed. In MAM4 this threshold is set to a relatively large

value. This is done to prevent BC from being removed too quickly by wet deposition—because the primary carbon mode has a lower hygroscopicity than the accumulation mode and thus a lower wet scavenging efficiency—thereby counteracting a low bias in BC concentrations in the Arctic regions. From Wang et al. (2018) we already know that using such a high threshold may not be appropriate. However, the global model also has biases in other processes that contribute to the low BC bias in the Arctic, and setting the threshold to a high value compensates for these errors. Our results are a reflection of this fact.

Figure 6g–l shows conditions representative of low latitudes. A grid cell sampled from the CESM2/MAM4 simulation (20°N, 120° E) contains 11% BC and 24% POM, distributed over the accumulation and primary carbon mode as shown in Figure 6g and j, with most of the mass in the accumulation mode. The corresponding value for $\chi_o$ is therefore 99%, an almost complete internal mixture. For the same reason, $\chi_c$ is also very high. Similarly to the high latitude case, Figure 6i and l shows that the comparable PartMC population has comparatively more BC-free particles and more particles that contain very low

amounts of primary carbonaceous material, leading to lower values of both $\chi_o$ and $\chi_c$ compared to the MAM4 results.

## 5.3 Comparison to observational data

The question that arises from Sections 5.1 and 5.2 is of course: Which spatial distribution of aerosol mixing state reflects reality more closely? The validation of simulated mixing state indices with observational data is still challenging since per-particle mass fractions of species are required for calculating the mixing state indices (see Section 3.1). These are in principle

obtainable from in situ deployments of single-particle mass spectrometers or by using electron microscopy techniques, but their quantitative derivation comes with challenges and is not routinely done, so that only very few datasets exist that allow for a meaningful comparison (Riemer et al., 2019). Keeping these limitations in mind, Zheng et al. (2021) reported a qualitative comparison of available measurements of mixing state metrics in locations in developed countries (Paris, France; Pittsburgh, USA; various locations in Japan) (Healy et al., 2014; Ye et al., 2018; Ching et al., 2019) with seasonally-averaged results from

the ML model based on particle-resolved simulations. This showed that the ML model was able to capture the range of values that is consistent with the observations.

We further compared the ML model estimates using recent observations from China. Specifically, we compared $\chi_o^{ML}$ and $\chi_o^{MAM4}$ with $\chi$ values from Taizhou (Zhao et al., 2021) and Beijing (Yu et al., 2020) derived from Single Particle Soot Photometer measurements. For both locations, $\chi_o^{MAM4}$ overestimated the observed $\chi$ values, while $\chi_o^{ML}$ was in the range of the

305 observations. Specifically, the $\chi$ measured at a suburban site Taizhou from 26 May to 18 June 2017 ranged from 62% to 82%. During the same time period (but in the year 2011), the values of $\chi_o^{ML}$ were between 63% and 84%, while $\chi_o^{MAM4}$ was between 84% and 96%. The $\chi$ values at the urban site of Beijing ranged between 55% and 70% in winter (from 10 November to 10 December 2016) and varied between 60% and 75% in summer (from 18 May to 25 June 2017). Using our simulations of the year 2011, $\chi_o^{ML}$ varied from 60% to 88% in winter and from 59% to 83% in summer. As a comparison, $\chi_o^{MAM4}$ ranged from

310 92% to 97% in winter and from 87% to 95% in summer. A caveat when comparing $\chi_o^{ML}$ and $\chi_o^{MAM4}$, respectively, with the observations reported in Zhao et al. (2021) and Yu et al. (2020) is that the definition of $\chi_o^{ML}$ and $\chi_o^{MAM4}$ included BC-free particles, while the $\chi$ values in the measurements by Zhao et al. (2021) and Yu et al. (2020) were calculated only considering





**Figure 6.** Illustration to explain the differences in mixing state representation between MAM4 and the ML model at high and low latitudes.





the subpopulation of BC-containing particles. This might introduce a bias in the mixing state index between the $\chi_o$ index used in this paper and the observations (depending on the fraction of the BC-free particles present at any given location).

We can also relate our $\chi_o$ index qualitatively to the Single Particle Soot Photometer measurements in the Finnish Arctic during winter 2011–2012 (Raatikainen et al., 2015). Although this study did not provide quantitative mixing state index calculations, it is an important finding that BC-containing particles (with various amounts of coatings) co-existed with BC-free particles. As we saw in Section 5.2, this condition can easily be represented with a particle-resolved approach. However, the modal model with modes configured as in MAM4 puts black carbon in all accumulation-sized particles (Figure 6), which is

not consistent with the observations.

## 6   Conclusions

In this paper we present a framework for evaluating the error in submicron aerosol mixing state induced by aerosol representation assumptions, which is one of the important contributors to structural uncertainty in aerosol models. We quantitatively compared mixing state indices for submicron aerosol predicted by the modal model MAM4 within the global model CESM

to a machine-learned model based on high-detail particle-resolved simulations. We focused on the mixing of optically absorbing and non-absorbing species ($\chi_o$), the mixing of primary carbonaceous with other aerosol species ($\chi_c$), and the mixing of hygroscopic and non-hygroscopic species ($\chi_h$).

For $\chi_o$, the MAM4 modal representation generally overestimated the degree of mixing of BC with other aerosol species. This overestimation is due to the fact that MAM4's choice of modes does not allow for representing BC-free particles in the

accumulation and primary carbon modes. This is in contrast to field observations by Brown et al. (2021), which showed that BC and POM may be externally mixed near sources. The implication of this is that, if optical properties are calculated based on the aerosol composition, absorption will be overestimated.

For $\chi_c$ and $\chi_h$, the error tended to be zonally structured, where the MAM4 model overestimated the mixing state indices at low latitudes, and underestimated them at high latitudes, compared to the ML model. This behavior could be explained

by modeling choices in MAM4, in particular that (1) BC is always emitted with POM, (2) no BC-free particles exist in the submicron modes, and (3) dust is always internally mixed with other aerosol species.

Mixing state is an important emergent property that affects the aerosol radiative forcing and aerosol-cloud interactions, but it is not easy to constrain this property globally. To the best of our knowledge, this is the first study that evaluated the spatial distribution of aerosol mixing state as predicted by a global model. Since errors in mixing state predictions propagate into

errors in aerosol climate impacts, our findings provide a framework and reference for Earth system model developers and users regarding simulation reliability. For example, this framework can be used to (1) quantify model bias in simulating mixing state in different regions, identifying model structural deficiencies, and (2) provide insights into potential improvements of model process representations for a more realistic simulation of aerosols.





**Table A1.** Aerosol mass and mass fraction definition and notation. The number of modes is $M$ ($M = 3$ for MAM4 without the coarse mode), the number of particles in mode $m$ is $N_m$, and the number of species is $A$.

| Quantity | Meaning |
|---|---|
| $\mu_{m,i}^a$ | mass of species $a$ in particle $i$ from mode $m$ |
| $\mu_{m,i} = \sum_{a=1}^A \mu_{m,i}^a$ | total mass of particle $i$ from mode $m$ |
| $\mu_m^a = \sum_{i=1}^{N_m} \mu_{m,i}^a$ | total mass of species $a$ from mode $m$ |
| $\mu_m = \sum_{i=1}^{N_m} \sum_{a=1}^A \mu_{m,i}^a$ | total mass of mode $m$ |
| $\mu^a = \sum_{m=1}^M \sum_{i=1}^{N_m} \mu_{m,i}^a$ | total mass of species $a$ in population |
| $\mu = \sum_{m=1}^M \sum_{i=1}^{N_m} \sum_{a=1}^A \mu_{m,i}^a$ | total mass of the population |
| $p_{m,i}^a = \frac{\mu_{m,i}^a}{\mu_{m,i}}$ | mass fraction of species $a$ in particle $i$ (within mode $m$) |
| $p_m^a = \frac{\mu_m^a}{\mu_m}$ | mass fraction of species $a$ in mode $m$ |
| $p_{m,i} = \frac{\mu_{m,i}}{\mu}$ | mass fraction of particle $i$ from mode $m$ in population |
| $p_m = \frac{\mu_m}{\mu}$ | mass fraction of mode $m$ in population |
| $p^a = \frac{\mu^a}{\mu}$ | mass fraction of species $a$ in population |

*Code and data availability.* Notebooks and data to reproduce the global mixing state indices analysis are available at https://github.com/
zzheng93/code_ms_ml_mam4 or https://doi.org/10.5281/zenodo.4731385.

**Appendix A:  Derivation of mode-based aerosol mixing state index**

Table A1 details the notation for aerosol mass and mass fractions to calculate $H_\alpha$ using modal information.

To explain how to obtain Eqs. (7) and (8) from Eqs. (2) and (3), let us assume that each mode $m$ contains $N_m$ particles, and the number of species in the population is $A$. The mixing entropy of particle $i$ in mode $m$, $H_{m,i}$, is given by

$$350 \quad H_{m,i} = \sum_{a=1}^A -p_{m,i}^a \ln p_{m,i}^a. \tag{A1}$$

The average particle mixing entropy of the entire population (summed over all modes), $H_\alpha$, is

$$H_\alpha = \sum_{m=1}^M \sum_{i=1}^{N_m} p_{m,i} H_{m,i} = \underbrace{p_{1,1} H_{1,1} + p_{1,2} H_{1,2} \cdots + p_{1,N_1} H_{1,N_1}}_{m=1} + \cdots + \underbrace{p_{M,1} H_{M,1} + p_{M,2} H_{M,2} \cdots + p_{M,N_M} H_{M,N_M}}_{m=M}. \tag{A2}$$

Given that each mode is assumed to be internally mixed, particles within the same mode have the same composition, and we have

$$355 \quad p_{m,i}^a = \frac{\mu_{m,i}^a}{\mu_{m,i}} = \frac{\mu_m^a}{\mu_m} = p_m^a. \tag{A3}$$


This results in

$$H_{m,i} = \sum_{a=1}^{A} -p_{m,i}^a \ln p_{m,i}^a = \sum_{a=1}^{A} -p_m^a \ln p_m^a = H_m. \tag{A4}$$

Therefore, based on Equation (A4) and the fact that $p_m = \sum_{i=1}^{N_m} p_{m,i}$, Equation (A2) can be rewritten as

$$H_\alpha = \underbrace{p_1 H_1}_{m=1} + \cdots + \underbrace{p_M H_M}_{m=M} = \sum_{m=1}^{M} p_m H_m. \tag{A5}$$

With the mode-based $H_\alpha$, the other mixing state quantities can be computed as described in Section 3.2.

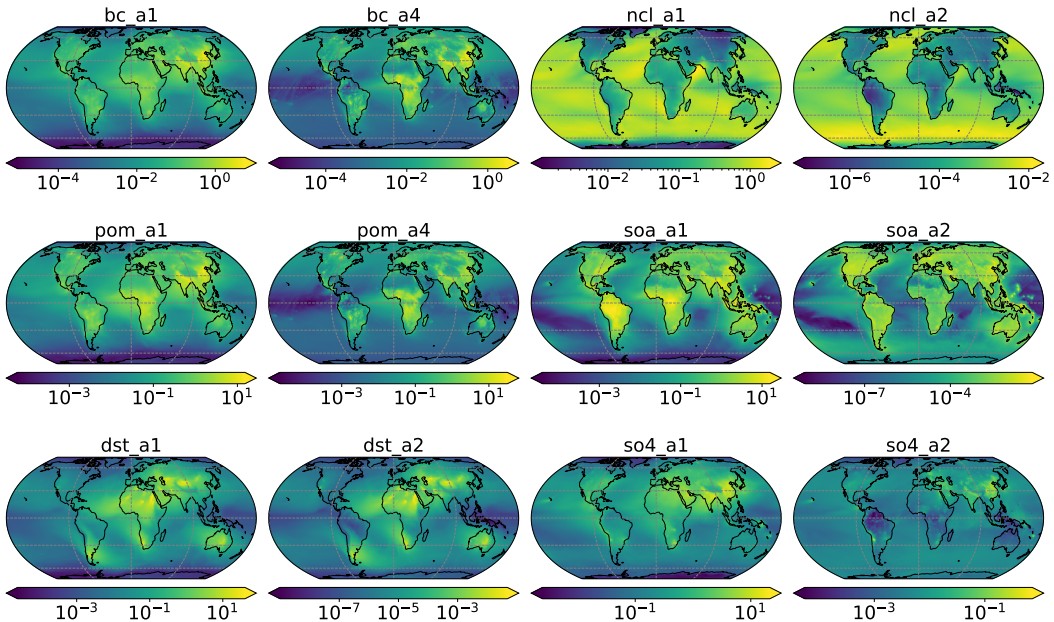

**Figure A1.** Aerosol species mixing ratio ($\mu$g/kg). a1: accumulation mode, a2: Aitken mode, a4: primary carbon mode, bc: black carbon, dst: dust, ncl: sea salt, pom: primary organic matter, soa: secondary organic aerosol, so4: sulfate.

*Author contributions.* ZZ, MW and NR conceptualized the analysis and wrote the manuscript with input from the co-authors. ZZ developed the code, carried out the simulations, and performed the analysis. LZ, PM, and XL provided scientific suggestions for the manuscript. All authors were involved in helpful discussions and contributed to the manuscript.

*Competing interests.* The authors declare that they have no conflict of interest.

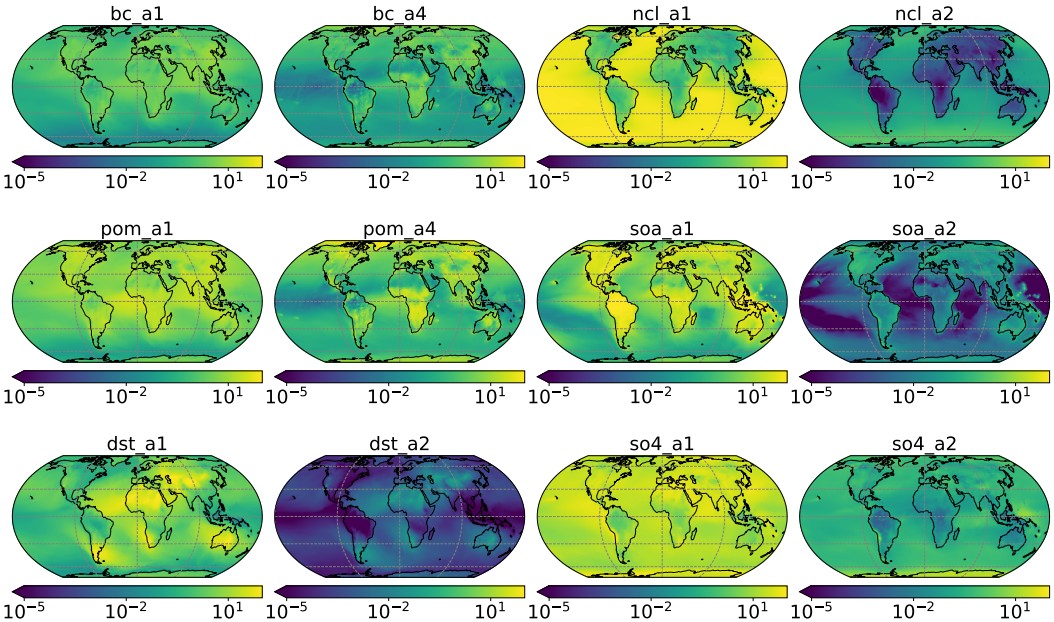

**Figure A2.** Fraction of aerosol species mixing ratio (%). a1: accumulation mode, a2: Aitken mode, a4: primary carbon mode, bc: black carbon, dst: dust, ncl: sea salt, pom: primary organic matter, soa: secondary organic aerosol, so4: sulfate.

*Acknowledgements.* We would like to acknowledge high-performance computing support from Cheyenne (doi:10.5065/D6RX99HX) provided by NCAR's Computational and Information Systems Laboratory, sponsored by the National Science Foundation. The CESM project is supported primarily by the National Science Foundation. This research is part of the Blue Waters sustained-petascale computing project, which is supported by the National Science Foundation (awards OCI-0725070 and ACI-1238993) the State of Illinois, and as of December, 2019, the National Geospatial-Intelligence Agency. Blue Waters is a joint effort of the University of Illinois at Urbana-Champaign

and its National Center for Supercomputing Applications. We also acknowledge funding from DOE grant DE-SC0019192 and NSF grant AGS-1254428. P.-L. Ma and X. Liu were supported by the "Enabling Aerosol-cloud interactions at GLobal convection-permitting scalES (EAGLES)" project (74358), funded by the U.S. Department of Energy, Office of Science, Office of Biological and Environmental Research, Earth System Model Development program. The Pacific Northwest National Laboratory is operated for the U.S. Department of Energy by Battelle Memorial Institute under contract DE-AC05-76RL01830.



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
