# Peer review of "Quantifying the structural uncertainty of the aerosol mixing state representation in a modal model"

_Atmospheric Chemistry and Physics, 2021_

## Author Comment (AC1)

**Response to the comments of Anonymous Referee #1 (RC2)**

Changes in response to the comments are marked in blue in the revised manuscript.

**(1.0)** The authors verified the global distribution of aerosol mixing state represented by modal models, focusing on comparing the calculations from ML model with MAM4. The author concluded the current model simulations on aerosol mixing state had large and zonally structured errors, and ML model trained on high-detail particle resolved simulations were more representative for realistic aerosols. The technical analysis is of reasonably high quality. The interpretation and discussion could benefit from some stronger quantitative analysis. The content is suitable for publication within the scope of ACP, while some revisions are required. Please see detailed comments below.

> *Thank you. We sincerely appreciate the Referee's comments, which helped us improve the quality of our manuscript.*

**(1.1)** Although ML model trained on high-detail particle resolved simulations is better than MAM4 for simulating aerosol mixing state, it still has some uncertainties which have not mentioned in this work. Taking the mixing of optically absorbing and non-absorbing species ($\chi_o$) for example, the ML simulations in this work are in the range of 50-90%, significantly higher than the realistic aerosols due to greatly underestimate of BC-free particles in some regions. Earlier work (Figure 8 in Atmos. Chem. Phys. Discuss., doi:10.5194/acp-2017-222, 2017) reported that 85%-90% of particles in the accumulation mode were BC-free particles, and internally mixed BC only accounted for 5-10% during summertime in North China Plain. The significantly underestimate of BC-free particles in ML simulations in this work is most likely attributed to irrespective of the processes of new particle formation and growth from Aitken mode to accumulation mode, which are hardly associated with particles (i.e., BC and POM) from primary emission. In term of the mixing of primary carbonaceous and non-primary carbonaceous species ($\chi_c$), the processes of new particle formation and growth will cause a similar uncertainty like the $\chi_o$. Therefore, in the regions where particles are dominated by new particle formation and growth processes, the methods used in the ML simulations in this work should be improved. The author should clarify this uncertainty as discussed above and give some suggests how to improve their ML simulations.

> *Thanks for raising this point. The reviewer is correct that neglecting the presence of particles that originate from new particle formation and growth from Aitken mode to accumulation mode will lead to an underestimation of BC-free particles, which in turn will lead to an overestimation of $\chi_o$, meaning that the overestimation of $\chi_o$ seen in MAM4 is even larger.*

> *However, we did consider new particle formation in some of the training data by directly introducing Aitken mode particles into the simulation (rather than explicitly simulating the process of new particle formation and growth explicitly, see Zheng et al. (2021)). The reason for this simplified treatment was that, while PartMC-MOSAIC includes the process of new particle formation (Tian et al., 2014), considerable uncertainty exists regarding the subsequent growth of the freshly nucleated particles (Kulmala et al., 2014), which poses a challenge for a highly detailed aerosol model such as PartMC-MOSAIC. We acknowledge that this approach may introduce uncertainties to the training dataset.*

> *To address the reviewer's concern, we have provided this information and added the clarification in the text (Line 173–180): "We note that new particle formation and growth was not simulated explicitly, but Aitken mode sulfate particles were introduced into the simulation by emission for a subset of scenarios (Zheng et al., 2021) as a proxy for having particles present that originate from new particle formation. While PartMC-MOSAIC includes the process of new particle formation (Tian et al., 2014), the reason for this simplfication was that considerable uncertainty exists regarding the subsequent growth of the freshly nucleated particles (Kulmala et al., 2014), which poses a challenge for a highly detailed aerosol model such as PartMC-MOSAIC. Errors in representing this particle type adequately may result in underestimating the abundance of BC-free*

*particles in some regions (Zhang et al., 2017) and thereby overestimating the degree of internal mixture. This would imply that the error in the MAM4 simulations is even larger than currently indicated."*

**(1.2)** Page 1/Line 7: The abbreviation can not be used here. Please give the full name of MAM 4.

*Changed to "4-mode version of the Modal Aerosol Module (MAM4)."*

**(1.3)** Page 4/Line 88: Why the authors ran the model for the year 2011 rather than more recent years?

*The period of the Community Atmosphere Model (CAM) scientifically supported compset "FHIST" ranges between 1979 and 2015 (please see the link). We wanted to take a year that is towards the end of the supported range, but not the very last year, and so we settled on 2011 (the fifth-last year). Other than this, the particular year does not have any significance for our study.*

**(1.4)** Page 5/Figure 1: In terms of the size distribution of the four modes (i.e., left panel), please show the diameter values of x-axis. If so, the readers can clear know the size ranges of the Aitken, accumulation, primary carbon and coarse modes.

*We updated this figure according to the reviewer's suggestion and also revised the placement of the modes according to the size ranges used in MAM4.*

[Figure]

Figure 1: Illustration of the mode-based calculation of the aerosol mixing state index. The coarse mode is removed because only modes dominated by submicron particles are used for calculations. Note that the Aitken mode mass fraction is very low compared to the other modes and the caption does not obscure any data.

**(1.5)** Page 6/Line 133: not "rather then", here should be "rather than".

*Done.*

**(1.6)** Page 6/Line 140: using a terminology as "mass absorption cross section" rather than "a mass absorption coefficient".

*This refers to the volume extinction coefficient divided by the mass concentration of BC, and is usually referred to as mass absorption coefficient (Petty, 2006, e.g.), so we left this as is.*

**(1.7)** Page 7/Line 164 and Line 176: For the input parameters of primary emissions of gas phase, the authors consider the NO2. Why not considering the nitrate in the aerosol species?

*Nitrate is not treated and thus not available in MAM4 ("FHIST" compset).*

**(1.8)** Page 7/Line 182: Please claim how to calculate the errors, namely 5% for $\chi_o$ and 8% for $\chi_c$ and $\chi_h$. What are the sources of the error? Why so small errors for model simulations?

*The "error" is the difference between PartMC-MOSAIC hold-out testing samples and the ML model predictions. This measures the emulation performance of the ML model (please see the caption of Table 2). Smaller errors corresponds to better ML model performance. The error comes from "bias error" and "variance", known as "bias–variance dilemma." In statistics and Machine Learning, the "bias error" refers to the error from assumptions in the learning algorithm, and the "variance" is the error from sensitivity to fluctuations in the training data set.*

*We have modified the text in Line 199–200: "The average error of the ML model (using the hold-out testing samples) is about 5% for $\chi_o$ and 8% for $\chi_c$ and $\chi_h$ (measured by mean absolute error)."*

**(1.9)** Page 11/Line 248–250: The authors claim that their simulations of ML model can not well represent the mixing state of sea salt. This will cause how much errors in the ocean regions?

*We did not mean to claim that our simulations of ML model cannot well represent the mixing state of sea salt. On the contrary, the good model performance (5% error for $\chi_o$ and 8% error for $\chi_c$ and $\chi_h$) could not be achieved if sea salt was not well represented.*

*We wanted to highlight that we cannot separate the impact induced by sea salt (also soa and so4) by comparing the difference in mixing state indices based on (pom, dst, ncl, soa, so4), (dst, ncl, soa, so4), and (ncl, soa, so4), since all three definitions contain sea salt, soa, and so4 in the same group.*

*Simply speaking, given the difference between (a+b+c) and (a+b), we can only determine the impact induced by c, but cannot sepearate the impacts of a and b individually.*

*We have rewritten the sentence in Line 268–269: "Based on our mixing state definitions, sea salt, soa and sulfate are always grouped together."*

**(1.10)** Page 13/Line 279–284: The authors state that the threshold is set to a relatively large value in MAM4 and their results are a reflection of this fact. Could the authors provide a reasonable value or range of the threshold based on their results?

*While adjusting the threshold criteria in MAM4 to a lower value may improve the agreement with the ML simulations in some regions, it may deteriorate the overall results in other areas. In fact, this is a great example of structural uncertainty, being intrinsic to the structure of the model formulation. It cannot be fixed by adjusting a parameter.*

*We added a sentence to the discussion in Line 304–307: "While adjusting the threshold criteria in MAM4 to a lower value may improve the agreement with the ML simulations in some regions, it may deteriorate the overall results in other areas. This is a good example how structural uncertainty manifests itself, namely by the fact that adjusting a parameter does not fundamentally fix the issue."*

**(1.11)** Page 15/Line 326–327: Here do not need the full name of $\chi_o$, $\chi_c$ and $\chi_h$.

*We think restating the full name of $\chi_o$, $\chi_c$ and $\chi_h$ in the conclusions would be helpful for readers to be reminded of the definitions. We therefore decided to leave this part of the manuscript as is.*

**(1.12)** Page 17-18/Figure A1 and Figure A2: The authors use a1, a2 and a4 represent different modes? Why is there no a3?

*Yes, "a3" (coarse mode) is not considered in this study. We added this to the captions of Figure A1 and Figure A2: "a1: accumulation mode, a2: Aitken mode, a4: primary carbon mode. The coarse mode (a3) is not used in this study and therefore omitted in the figure".*

**Response to the comments of Anonymous Referee #2 (RC3)**

Changes in response to the comments are marked in blue in the revised manuscript.

**(2.0)** Zheng and coauthors have undertaken a novel comparison of predicted aerosol mixing state between their highly-detailed particle-resolved model and an existing Earth system model (CESM2). The authors are able to leverage the impressive numerical formulation of PartMC-MOSAIC and the machine learning approaches they have developed to provide some context as to how well the widely used modal method is able to capture aerosol mixing state. The paper is especially well-written, the content is well-organized, figures are well-constructed. In short, the manuscript was a pleasure to read. I encourage ACP to publish this paper, considering it is one of many steps toward a more sophisticated future landscape of aerosol modeling approaches. However, I do have a couple of topics I would appreciate the authors commenting on in the manuscript to make sure they are resolved or documented before moving on from this study.

> *We thank the Referee for the comments on the contribution of our study.*

**(2.1)** The relevance of the selected ML input parameters for predicting aerosol mixing state is clear, especially for an exercise where one is feeding the inputs from actual measurements to model. However, in this study, it is the CESM2 fields that are being used to specify the parameters, including the concentrations of each aerosol species. Since the aerosol formulation in the CESM2 has an impact on the speciated aerosol concentrations (e.g. the authors mention the example of BC transferring to the mixed mode and depositing faster), how can the authors be confident that the ML fields in Fig. 3 are actually realistic if they are based on CESM2 model data that they are arguing is at least partially corrupted? Forgive the unproven hypothetical here, but in the extreme, couldn't it be possible that the MAM4 model gives erroneous PM species concentrations that compensate with an erroneous framework to yield an accurate mixing ratio field? The authors provide helpful discussion in section 5.3 to address this, so that extreme example is unlikely. But would they consider adding some results that show the variability in their ML fields with reasonable variation in the underlying inputs, perhaps based for input ranges informed by the biases of the CESM2 speciated PM predictions?

> *The reviewer is completely correct: If CESM2 gives erroneous PM species concentrations, our ML framework has no means to correct for this. Instead, what we can expect is that it provides the "most likely" mixing state associated with the species concentrations that CESM2 simulates.*
>
> *We added the following text in Line 201–203: "We would like to emphasize that this ML modeling framework cannot compensate for any biases that the global model (here CESM2) might have in simulating the quantities that serve as the features. Instead, what we can expect from this approach is that it provides the "most likely" mixing state associated with the species concentrations that CESM2 simulates."*

**(2.2)** Additionally, would the authors please explain why other parameters relevant for sources are not appropriate for the ML input? I'm thinking of data like land use, human population, and wind speed that could help inform dust, carbon, and sea spray mixing state. If it's been documented, as the authors say, that the mixing state parameter goes down close to sources, why not use that information in this exercise? Maybe this would require running the ML training cases for longer than 24 hrs to get a signal for aging (i.e. distance from sources) on mixing state?

> *Other features (e.g., land use, human population) are not available in both PartMC-MOSAIC and CESM2 simulations. Additional features could be helpful if they were available.*
>
> *We added the following sentence in the paper (Line 191–192): "The choice of features is determined by the overlap of variables that are present in both PartMC-MOSAIC and CESM2."*

**(2.3)** My main concern with this study is that the authors have chosen to focus on surface data from the global model, and yet are interested in impacts on radiative forcing and cloud-aerosol interactions. Certainly, to

know the true scope of the potential problem in misrepresentation of aerosol mixing state, the full atmospheric column needs to be taken into account.

*Yes, we agree with the Referee. The current PartMC box model simulations were designed to represent aerosol processes in the boundary layer, and therefore we limit our analysis to layers near the surface. Our ultimate goal is to extend this to the full atmospheric column, which needs 3-D WRF-PartMC simulations and is currently in the works.*

**(2.4)** Do the authors have any idea how PartMC-MOSAIC ML performs with scavenging in place? The ML input parameters don't include hydrometeors or cloud cover as a marker for aerosol removal. It seems like the impact of a major process like cloud scavenging will be extremely important to understand from at least a few points of view: 1) impact on ML representation of PartMC-MOSAIC and 2) divergence between ML and MAM4 in the context of clouds/fog.

*Another great question! For the purpose of this study, the important thing is that we (somehow) generate a sufficiently comprehensive set of aerosol populations that can serve as training data. The safest way to accomplish this is to indeed generate these populations by including all possible processes that aerosols undergo in the real atmosphere. This is in practice very challenging, and so we rely on the assumption that our training data set is "good enough" even though we do not explicitly include the impact of scavenging.*

*We included a sentence in the paper to add this as an explicit caveat (in Line 180—183): "Other processes that are not explicitly included in generating training data are aerosol removal by nucleation-scavenging and other cloud processes. However, for the purpose of this study, the emphasis is on the aerosol state, i.e., having a sufficiently comprehensive set of aerosol populations that can serve as training data, not necessarily that all the processes are included."*

**(2.5)** Abstract: I recommend adding a few words or a sentence to the end of the abstract that explicitly connects back to the main motivation, which I assume to be a better quantification of radiative forcing and aerosol-cloud interactions. How significant is the 70% modal model bias for those aims?

*The reviewer is correct that this is the main motivation. We updated the abstract as follows: "Our study quantifies potential model bias in simulating mixing state in different regions, and provides insights into potential improvements to model process representation for a more realistic simulation of aerosols towards better quantification of radiative forcing and aerosol-cloud interactions."*

*Regarding the significance of the 70% bias, there are several steps to get there. The first is to figure out how to compare "mixing state" as it is predicted by different models. This is the focus of this paper. The second step is to figure out how bias in mixing state translates into bias in radiative forcing. At this time we don't have a pipeline set up to evaluate this and so we need to leave this to further research.*

**(2.6)** Page 2, lines 38–43: A related issue is the transfer of particles from smaller modes to larger ones during growth, even if this is among "pure" modes. Also, removal of particles due to scavenging by cloud activation is difficult to reconcile.

*Thank you. We have added the points into Line 41–43: "Generally, the transfer of aerosol mass from smaller modes to larger modes during growth can lead to inaccuracies. The removal of particles due to scavenging by cloud activation is another issue that is difficult to reconcile."*

**(2.7)** Page 4, line 88: Please add an explicit reference for the MAM4 long-range transport bias.

*We added a reference to Liu et al. (2016) in Line 91.*

**(2.8)** Fig. 1: Legend obscures data in right panel.

*The Aitken mode mass fraction in this figure is very low compared to other modes. We did not find that the legend obscures data in the right panel in Fig. 1. We added a sentence to the caption to clarify this: "Note that the Aitken mode mass fraction is very low compared to the other modes and the caption does not obscure any data." (Please see our response to comment (1.4))*

**(2.9)** Section 3.3: I'm sure there are many applications for grouping surrogate species in different ways, but I think an important one worth mentioning is heterogeneous reactions on surfaces of (e.g.) primary dust or sea spray particles. There is probably much overlap with the hygroscopicity mixing state index. If the authors agree, perhaps a comment could be added to that effect.

*This is a good idea. We modified the sentence (Line 119–123): "Note that the definition of "species" for calculating $\chi$ is based on application needs. It can be based on operationally defined chemical species (Healy et al., 2014; Ye et al., 2018), elemental composition (Fraund et al., 2017; O'Brien et al., 2015), or species groups such as volatile and nonvolatile species (Dickau et al., 2016) or hygroscopic and non-hygroscopic species (Ching et al., 2017; Hughes et al., 2018). Other possibilities include the prospensity for aerosols to undergo heterogeneous reactions, quantified by the heterogeneous reaction rate coefficient for a specific reaction."*

**Response to the comments of Anonymous Referee #3 (RC1)**

Changes in response to the comments are marked in blue in the revised manuscript.

**(3.0)** The current work evaluated the several mixing state indices ($\chi$s) derived from a global modal model (CESM2/MAM4) by a global distribution of $\chi$s derived from the machine learning (ML) model based on the results of the mixing state resolving model PartMC/MOSAIC. The authors also compared their $\chi$s against those obtained from the field observation data. This is the first study to evaluate the spatial distribution of $\chi$s in models, which are currently used for the climate predictions. Let me congratulate the authors to achieve this. The manuscript will be acceptable after the authors address the following couple of minor comments, general and specific ones.

> *Thank you. We appreciate the Referee's comments on the contribution of our study.*

**(3.1.1)** The differences of $\chi$s between CESM2/MAM4 and PartMC/MOSAIC-ML should originate from different aerosol representations and different rates in aerosol dynamical processes such as rates of secondary aerosol formations, condensation/evaporation, and coagulation. However, the authors only addressed the former, the difference in aerosol representations. Is it because the latter (difference in aerosol dynamics processes) is negligibly small with compared to the former (difference in aerosol representations)? If so, please describe how the authors quantified or estimated them.

> *Our usage of the term "aerosol representation" in this paper encompasses the representation of processes that go along with the aerosol representation itself, since the two are in practice tightly coupled). We added this sentence in Line 51–52.*

**(3.1.2)** Are the condensation/coagulation rates or size distributions of aerosols of the two models similar with each other?

> *Yes, these rates are similar in PartMC-MOSAIC and in CESM2 when the underlying populations are similar. We matched size distributions to make sure that the training data encompasses what is found in the global model (Zheng et al., 2021).*

**(3.2)** How the existence of coarse mode particles in CESM2/MAM4 can make the global $\chi$s distributions different from those of PartMC/MOSAIC-ML which does not consider the existence of coarse mode particles? A part of secondary gases condenses to coarse mode particles such as mineral/anthropogenic dust and sea salt, and thus neglection of coarse mode particles can cause overestimation of condensational growth of fine mode particles. Is this effect negligibly small in the current assessment?

> *We would like to clarify that although the mixing state calculation did not take the coarse mode into account, the coarse mode particles were present throughout the PartMC-MOSAIC simulations. Therefore it would not be the source of errors.*
>
> *We added the following sentence in the paper (Line 167–170): "The strategy to generate the data was to vary the input parameters (45 in total) for the PartMC-MOSAIC model, including primary emissions of different aerosol types (e.g., carbonaceous aerosol and dust emissions, including contribution from Aitken mode, accumulation mode, and coarse mode size ranges), primary emissions of gas phase species (e.g., SO2, NO2, and various volatile organic compounds), and meteorological parameters (see Table 1 in Zheng et al. (2021) for more information)."*

**(3.3)** Uppercase letters are used for the abbreviations of aerosol species in Sects. 2 and 3, such as SOA, POM, and BC, while lowercase letters (pom, dst, ncl, soa, and so4) are used in the panels. Any reasons? "soa" in panels and tables are different in definitions from SOA in the main text?

*The lowercase letters are used to be consistent with the CESM field names, while the uppercase letters are the common expressions. They refer to the same things.*

**(3.4)** Figure 1 is not referred in the main text.

*Thank you. We have modified the text (Line 126) "The framework laid out in Section 3.1 can be easily generalized to a modal modeling framework (see Figure 1)".*

**References**

Ching, J., Fast, J., West, M., and Riemer, N.: Metrics to Quantify the Importance of Mixing State for CCN Activity, Atmos. Chem. Phys., 17, 7445–7458, https://doi.org/10.5194/acp-17-7445-2017, 2017.

Dickau, M., Olfert, J., Stettler, M. E. J., Boies, A., Momenimovahed, A., Thomson, K., Smallwood, G., and Johnson, M.: Methodology for Quantifying the Volatile Mixing State of an Aerosol, Aerosol Sci. Technol., 50, 759–772, https://doi.org/10.1080/02786826.2016.1185509, 2016.

Fraund, M., Pham, D., Bonanno, D., Harder, T., Wang, B., Brito, J., de Sá, S., Carbone, S., China, S., Artaxo, P., Martin, S., Pöhlker, C., Andreae, M., Laskin, A., Gilles, M., and Moffet, R.: Elemental Mixing State of Aerosol Particles Collected in Central Amazonia during GoAmazon2014/15, Atmosphere, 8, 173, https://doi.org/10.3390/atmos8090173, 2017.

Healy, R. M., Riemer, N., Wenger, J. C., Murphy, M., West, M., Poulain, L., Wiedensohler, A., O'Connor, I. P., McGillicuddy, E., Sodeau, J. R., and Evans, G. J.: Single Particle Diversity and Mixing State Measurements, Atmos. Chem. Phys., 14, 6289–6299, https://doi.org/10.5194/acp-14-6289-2014, 2014.

Hughes, M., Kodros, J., Pierce, J., West, M., and Riemer, N.: Machine Learning to Predict the Global Distribution of Aerosol Mixing State Metrics, Atmosphere, 9, 15, https://doi.org/10.3390/atmos9010015, 2018.

Kulmala, M., Petäjä, T., Ehn, M., Thornton, J., Sipilä, M., Worsnop, D., and Kerminen, V.-M.: Chemistry of Atmospheric Nucleation: On the Recent Advances on Precursor Characterization and Atmospheric Cluster Composition in Connection with Atmospheric New Particle Formation, Annu. Rev. Phys. Chem., 65, 21–37, https://doi.org/10.1146/annurev-physchem-040412-110014, 2014.

Liu, X., Ma, P.-L., Wang, H., Tilmes, S., Singh, B., Easter, R. C., Ghan, S. J., and Rasch, P. J.: Description and Evaluation of a New Four-Mode Version of the Modal Aerosol Module (MAM4) within Version 5.3 of the Community Atmosphere Model, Geosci. Model Dev., 9, 505–522, https://doi.org/10.5194/gmd-9-505-2016, 2016.

O'Brien, R. E., Wang, B., Laskin, A., Riemer, N., West, M., Zhang, Q., Sun, Y., Yu, X.-Y., Alpert, P., Knopf, D. A., Gilles, M. K., and Moffet, R. C.: Chemical Imaging of Ambient Aerosol Particles: Observational Constraints on Mixing State Parameterization, J. Geophys. Res. Atmos., 120, 9591–9605, https://doi.org/10.1002/2015JD023480, 2015.

Petty, G.: A first course in atmospheric radiation, Sundog Pub., Madison, WI, 2nd edn., 2006.

Tian, J., Riemer, N., West, M., Pfaffenberger, L., Schlager, H., and Petzold, A.: Modeling the Evolution of Aerosol Particles in a Ship Plume Using PartMC-MOSAIC, Atmos. Chem. Phys., 14, 5327–5347, https://doi.org/10.5194/acp-14-5327-2014, 2014.

Ye, Q., Gu, P., Li, H. Z., Robinson, E. S., Lipsky, E., Kaltsonoudis, C., Lee, A. K., Apte, J. S., Robinson, A. L., Sullivan, R. C., Presto, A. A., and Donahue, N. M.: Spatial Variability of Sources and Mixing State of Atmospheric Particles in a Metropolitan Area, Environ. Sci. Technol., 52, 6807–6815, https://doi.org/10.1021/acs.est.8b01011, 2018.

Zhang, Y., Su, H., Kecorius, S., Wang, Z., Hu, M., Zhu, T., He, K., Wiedensohler, A., Zhang, Q., and Cheng, Y.: Mixing State of Refractory Black Carbon of the North China PlainRegional Aerosol Combining a Single Particle Soot Photometer Anda Volatility Tandem Differential Mobility Analyzer, Preprint, Aerosols/Field Measurements/Troposphere/Chemistry (chemical composition and reactions), https://doi.org/10.5194/acp-2017-222, 2017.

Zheng, Z., Curtis, J. H., Yao, Y., Gasparik, J. T., Anantharaj, V. G., Zhao, L., West, M., and Riemer, N.: Estimating Submicron Aerosol Mixing State at the Global Scale With Machine Learning and Earth System Modeling, Earth Space Sci, 8, https://doi.org/10.1029/2020EA001500, 2021.